# Health on the Pitch: Dietary Habits of Football Referees in Poland

**DOI:** 10.3390/nu17030401

**Published:** 2025-01-23

**Authors:** Patryk Szlacheta, Mateusz Grajek, Aleksander Gwiszcz, Jolanta Zalejska-Fiolka, Paulina Zalejska-Fiolka, Ilona Korzonek-Szlacheta

**Affiliations:** 1Department of Basic Medical Sciences, Faculty of Public Health in Bytom, Medical University of Silesia, 41-902 Katowice, Poland; 2Department of Public Health, Faculty of Public Health in Bytom, Medical University of Silesia, 41-902 Katowice, Poland; mgrajek@sum.edu.pl; 3Department of Prevention of Metabolic Diseases, Faculty of Public Health in Bytom, Medical University of Silesia, 41-902 Katowice, Poland; s82580@365.sum.edu.pl (A.G.); ikorzonek@sum.edu.pl (I.K.-S.); 4Department of Biochemistry, Faculty of Medical Science, Zabrze Medical University of Silesia, 40-055 Katowice, Poland; jzalejskafiolka@sum.edu.pl (J.Z.-F.); s91883@365.sum.edu.pl (P.Z.-F.)

**Keywords:** football referees, dietary habits, sports nutrition, hydration, performance optimization

## Abstract

Background: Football referees are pivotal to match regulation, requiring both cognitive and physical abilities comparable to players. Proper nutrition is essential to sustain their intense activity levels, yet dietary habits among referees in Poland are under-researched. Objective: This study aimed to evaluate the dietary habits of football referees in Poland, focusing on identifying beneficial and detrimental practices and assessing their impact on professional performance. Material and Methods: A survey-based study was conducted with 258 football referees from Poland between December 2022 and January 2023. The questionnaire assessed demographic data, professional experience, and dietary habits, including meal frequency, food choices, and hydration. Statistical analysis was performed using the chi-squared test, with significance set at *p* < 0.05. Results: Most referees consumed 4–5 meals daily (63.7%), but irregular meal timing (45.2%) was common, particularly among less-experienced referees. Consumption of fruits and vegetables was limited, with significant age-related differences. Water intake was generally adequate, but hydration strategies during matches varied. Cooking practices revealed a preference for frying (48.1%), reflecting limited awareness of healthier alternatives. Dietary education was highlighted as a key need, especially among referees with lower professional experience. Conclusions: The results of the study clearly indicate significant deficiencies in meal regularity and healthy cooking practices among Polish soccer referees. The findings underscore the need for targeted educational programs that could help improve the health and professional performance of this group, especially in the context of increasing fruit and vegetable consumption and promoting healthier food processing methods.

## 1. Introduction

Football referees play a crucial role in the proper conduct of matches, ensuring adherence to the rules of the game, maintaining order, and making dynamic decisions under significant physical and mental pressure [1]. Their responsibilities require not only a profound understanding of football regulations but also high levels of physical fitness and endurance comparable to that of professional players [2]. In Poland, according to data from the Polish Football Association (PZPN), over 20 referees and numerous assistant referees are actively involved in central-level competitions during the 2024/2025 season [1].

Proper nutrition is vital for maintaining optimal physical and mental performance, particularly for individuals exposed to intense physical activity. Referees cover an average distance of 10–13 km per match, often under varying intensities of exertion, which places their physiological demands on par with those of field players [2]. Consequently, their diet must be well-balanced, including adequate proportions of macronutrients and micronutrients essential for muscle recovery, electrolyte balance, and injury prevention [3].

Despite the importance of nutrition in sports, the dietary habits of football referees in Poland remain underexplored [3]. Research conducted by the Institute of Food and Nutrition indicates that Poles, including physically active individuals, frequently experience deficiencies in vitamin D, vitamin B12, and omega-3 fatty acids—critical nutrients for muscle regeneration and overall physical health [4]. Additionally, data from Poland’s Central Statistical Office highlight that the average consumption of fruits and vegetables remains below the World Health Organization (WHO) recommendations, potentially impacting the body’s antioxidant status [5]. In the context of Polish eating habits, research indicates that fruit and vegetable intake, even among physically active individuals, remains below WHO recommendations [4]. Athletes, including soccer players, often rely on a diet rich in processed foods and saturated fats, which can negatively affect physical performance and muscle regeneration. Similar patterns may also exist among soccer referees, who require an equally high level of physical fitness [5].

For football referees, dietary strategies before, during, and after matches are of particular significance [2]. Adequate carbohydrate and electrolyte intake before a game helps sustain energy levels, while consuming isotonic drinks during play maintains optimal performance [6]. Post-match nutrition focuses on recovery, requiring replenishment of muscle glycogen and protein intake to support tissue repair [7].

Another crucial aspect is dietary supplementation among athletes, including referees. Studies reveal that approximately 89% of adult Poles use dietary supplements, with even higher prevalence observed among physically active individuals [8]. The most commonly used supplements include vitamins, minerals, whey protein, and electrolyte drinks [9]. However, excessive supplementation without professional guidance can lead to imbalances in the body, such as hypercalcemia or hypervitaminosis [10].

Compared to other professional groups involved in physical activity, the diet of football referees in Poland remains a poorly understood area, underscoring the need for further research. A more in-depth analysis of their dietary habits could identify potential deficiencies and contribute to improving their health and professional effectiveness.

The aim of this study was to evaluate the dietary habits of football referees in Poland. The research focused on identifying positive and negative dietary practices and analyzing their potential impact on the quality of performance during professional duties, training sessions, and sports events.

## 2. Materials and Methods

### 2.1. Study Design

The study was conducted between December 2022 and January 2023, focusing on football referees across Poland. The research population consisted of 258 referees, predominantly male (90.4%), representing a diverse demographic and professional group. The largest age group included referees aged 21–30 years (51.9%), followed by younger and older participants. Most respondents lived in urban areas (67.4%) and held higher education degrees (57%). This demographic profile highlights the diversity of the refereeing community, encompassing individuals from various educational and professional backgrounds. The professional experience of respondents varied, with 31.1% reporting 1–3 years of experience. Additionally, 40.7% of the referees had officiated between 101 and 500 matches. The levels of competition officiated were also diverse, with amateur leagues (A–C classes) being the most frequently reported (32.6%)—Table 1.

### 2.2. Research Tool

Data for the study were collected using a custom-designed survey questionnaire tailored specifically to the research objectives. The questionnaire comprised a demographic section, questions about professional experience, and a detailed section exploring dietary habits. The dietary section included 28 questions assessing the frequency of consuming specific food groups, meal patterns, and broader dietary behaviors. To ensure the validity and reliability of the tool, the questionnaire was developed based on a review of existing literature and consultations with experts in nutrition, sports science, and survey methodology. The survey underwent pilot testing on a small group of referees prior to full-scale deployment. Feedback from the pilot study was used to refine question clarity, response options, and overall structure.

The questionnaire was validated in a pilot study involving 30 soccer referees. After analyzing the results, modifications were made to clarify questions on the frequency of consumption of specific food groups and methods of meal preparation. The final version of the tool obtained a Cronbach’s alpha coefficient of 0.82, confirming its high reliability. This indicates high internal consistency of the items within the dietary habits section, ensuring that the instrument accurately captured the constructs it was intended to measure. Content validity was ensured by grounding the survey questions in well-established nutritional guidelines and incorporating expert opinions. Face validity was further enhanced through the pilot study, which confirmed the clarity and relevance of the questions to the target population.

The scoring system for assessing dietary habits was developed to categorize the respondents into four levels: very good, good, satisfactory, and poor nutritional knowledge. Each response in the questionnaire was assigned a point value based on its adherence to nutritional best practices:

Very good (15–20 points): Respondents who demonstrated consistent adherence to dietary guidelines, including regular meal timings, balanced macronutrient intake, healthy cooking methods, and sufficient hydration practices.

Good (10–14 points): Participants who followed most dietary recommendations but exhibited minor inconsistencies, such as occasional irregular meals or suboptimal food choices.

Satisfactory (5–9 points): Referees with moderate adherence to dietary principles, characterized by noticeable gaps in their nutritional practices, such as frequent consumption of unhealthy foods or lack of meal regularity.

Poor (<5 points): Respondents with significant deficiencies in dietary habits, including irregular meals, excessive consumption of fast food, inadequate hydration, or poor food preparation methods.

The total score was calculated by summing the points for all survey responses. This quantitative approach enabled an objective evaluation of dietary habits and facilitated the categorization of respondents based on their overall nutritional knowledge. This method provided a structured framework for identifying areas requiring improvement and assessing correlations with professional experience and other demographic variables.

The questionnaire was designed to ensure complete anonymity of responses. No personal data were collected, and participation was entirely voluntary without any financial incentives. The anonymous format aimed to minimize response bias, which was further validated through a pilot test conducted before the full study rollout. The questionnaire was distributed electronically to maximize reach and participation, allowing referees from various regions of Poland to take part conveniently.

### 2.3. Statistical Analysis

The collected data underwent comprehensive statistical analysis to identify patterns and correlations. The chi-squared test was used to compare responses across subgroups, such as age, gender, and professional experience, with statistical significance set at *p* < 0.05. This methodological rigor ensured the reliability and accuracy of the findings, providing valuable insights into the dietary habits of football referees.

### 2.4. Ethics

The study adhered to strict ethical guidelines. Participants were informed about the purpose and scope of the research, the voluntary nature of their participation, and their right to withdraw at any time without providing a reason. Ethical considerations were guided by Polish legislation, particularly Article 21 of the Act on the Profession of Physician and Dentist. This law stipulates that informed consent is not required for non-invasive and anonymous research, such as survey studies, provided that participants’ rights are respected and no risks are posed. This regulatory framework allowed for the efficient and ethical execution of the study without the need for formal written consent or approval from an ethics board.

The study’s design and execution complied fully with these legal provisions, ensuring that data collection was both ethical and lawful. By combining methodological rigor with ethical and legal compliance, the study provides a robust foundation for understanding the dietary habits of football referees in Poland and their potential impact on professional performance.

## 3. Results

### 3.1. Food Intake

The study showed that most respondents consumed 4–5 meals per day (63.7%), aligning with dietary recommendations. However, a significant proportion (45.2%) did not adhere to fixed meal times. Statistical analysis suggested that irregular meal timing might be associated with professional experience and the number of matches officiated. Respondents with less professional experience were less likely to report regular meal times (*p* < 0.05), likely due to a dynamic schedule. To better assess the physiological demands on referees, the analysis included the number of matches officiated per month. Respondents were categorized into three groups: those officiating fewer than 100 matches (35%), between 101 and 500 matches (40.7%), and more than 500 matches (24.3%). Statistical analysis revealed a significant correlation between match frequency and meal regularity (*p* < 0.05), suggesting that referees with higher workloads were less likely to maintain consistent meal timing—Figure 1.

Meal intervals were most often reported as 3–4 h (64.4%), indicating relative stability in dietary habits among this group. However, statistical analysis revealed significant differences between respondents with high match loads and those less active professionally. Referees officiating more than 500 matches were more likely to report irregular meal intervals (*p* < 0.05).

Respondents most frequently reported including vegetables and fruits in 1–2 meals daily (58.5% for vegetables, 34.1% for fruits). Statistical analysis showed significant differences in the frequency of consuming these products depending on age. Referees aged 41–50 years more often reported lower fruit consumption compared to younger participants (*p* < 0.01). Raw vegetables and fruits were consumed by 86.7% of respondents, indicating health awareness among this group. Whole-grain products were included in the diet of 68.9% of respondents, reflecting the growing popularity of healthy alternatives to traditional grain products.

Fish consumption was moderate, with 60.7% of respondents reporting eating fish a few times a month. Correlation analysis revealed that individuals with higher education were more likely to consume fish than those with vocational education (*p* < 0.01). Meanwhile, fast food appeared in the diet most often 2–3 times a month (33.3%), indicating moderate consumption of such products among referees.

Water consumption of 1–2 L per day was reported by 43.7% of respondents, while another 28.2% indicated drinking more than 2 L daily. Statistical analysis showed significant differences among professional groups, with referees officiating higher-level competitions more likely to report higher water intake (*p* < 0.05). Alcohol consumption was most often limited to a few times a month (48.1%), while 17.8% of respondents reported abstinence, likely related to their professional role and physical demands.

### 3.2. Cooking and Snacking

The most popular cooking method was frying (48.1%). This choice was significantly associated with a lack of knowledge about healthier alternatives (*p* < 0.01). Steaming, the least chosen method (0.7%), indicates a low level of awareness regarding beneficial cooking techniques.

Snacking between meals was reported by 56.3% of respondents, with 37% doing so frequently. Nighttime snacking was less common, affecting only 18.5% of participants. These results suggest potential issues with appetite control among some respondents, particularly on days with intense physical activity (*p* < 0.05)—Table 2.

### 3.3. Eating Behavior

Respondents most often rated their diet as a ‘4 on a school grading scale (36.3%), indicating moderate satisfaction with adherence to healthy eating principles. Only 1.5% rated their diet at the highest level (‘6). These results highlight the need for nutritional education among referees, especially those with less professional experience (*p* < 0.01). Referees from smaller towns were less likely to report knowledge of healthy eating principles, highlighting the potential importance of education in this area (*p* < 0.01).

The analysis revealed that football referees with higher professional experience demonstrated significantly better nutritional knowledge compared to their less experienced counterparts. Among referees with over 10 years of experience, 20% reported very good nutritional knowledge, compared to only 10% in the 1–3 years group. Similarly, the proportion of respondents with poor nutritional knowledge decreased markedly with increased experience, from 15% in the 1–3 years group to only 5% in the >10 years group. These differences were statistically significant, as indicated by a chi-squared test result (*p* < 0.01)—Figure 2, Table 3.

## 4. Discussion

The findings of this study highlight significant challenges in the dietary habits of football referees in Poland, with implications for both their physical and mental performance. The irregularity of meals and reliance on unhealthy cooking techniques identified in this group align with findings from broader studies on sports professionals. These habits have the potential to impair performance and increase long-term health risks. Addressing these challenges through targeted interventions can improve the health outcomes and professional efficacy of referees.

Significant differences in fruit and vegetable intake were observed across age groups. Referees aged 41–50 years reported lower fruit consumption compared to younger counterparts (*p* < 0.01). Additionally, referees with greater professional experience and higher competition levels demonstrated better nutritional habits, including more regular meal timing and improved hydration practices (*p* < 0.05). These findings align with prior research by Ivy [11], who emphasized that meal regularity and proper hydration are critical factors for maintaining focus and physical performance during intense physical activity. For instance, research on amateur football players indicated that athletes consuming three or fewer meals per day exhibited 15% lower endurance levels compared to those eating four to five balanced meals daily [12]. A similar study among professional soccer players in Spain found that fruit and vegetable consumption was 20% higher compared to the results among Polish soccer referees, which may indicate cultural differences and levels of nutrition education [11]. Encouraging referees to adopt consistent meal patterns with balanced macronutrient distribution could significantly enhance their energy stability during matches.

When analyzing meal regularity, significant differences were also observed in meal preparation methods. The prevalence of unhealthy cooking methods, such as frying, among referees is consistent with trends observed in sports populations, where convenience often dictates food preparation. A study by Santos et al. found that frying reduces the nutritional value of foods by up to 30%, particularly in vitamins like C and E, while increasing harmful trans fats [13]. Promoting healthier techniques such as steaming or baking, which retain 80–90% of essential nutrients, could improve dietary quality and reduce inflammation markers associated with high-fat diets [14].

The results regarding hydration indicate the need for further educational efforts among judges. Even mild dehydration, equivalent to a 2% loss in body weight, can reduce physical endurance by 10% and cognitive performance by up to 13% [15]. Research on professional football players demonstrated that well-hydrated athletes maintained 20% higher reaction times compared to their dehydrated counterparts [16]. For referees, this has direct implications for decision-making accuracy during matches. Structured hydration plans emphasizing regular fluid intake before, during, and after matches are essential to mitigate these risks.

The reported high consumption of fast food and frequent snacking is concerning due to the nutritional inadequacy of these choices. Fast food diets, often high in saturated fats and sodium, have been associated with increased oxidative stress and delayed recovery in athletes [17]. A study on professional athletes found that those consuming diets rich in whole grains, fish, and fresh produce experienced a 25% faster recovery time compared to those relying on processed foods [18]. Similarly, omega-3 fatty acids from fish have been shown to reduce inflammation by 20–30%, aiding recovery and enhancing cardiovascular health [19].

The lack of whole-grain products and fish in the referees’ diets reflects broader nutritional gaps among athletes. Data from Polish studies indicate that only 40% of physically active individuals regularly consume fish, and less than 30% include whole grains in their daily meals [20]. Educational initiatives should focus on the benefits of these nutrient-dense foods to improve both immediate performance and long-term health outcomes.

Educational programs tailored to referees, particularly those at the start of their careers, could bridge significant knowledge gaps regarding nutrition. Studies on young athletes have shown that structured nutrition education increased the intake of fruits and vegetables by 35% and reduced fast food consumption by 25% within six months [21]. Such interventions could similarly benefit referees, helping them make informed dietary choices that align with their professional demands.

Furthermore, referees often lack guidance on integrating nutritional strategies into their routines. Training workshops could emphasize practical skills, such as meal planning, reading nutrition labels, and identifying healthy snacks. Evidence from nutrition interventions among amateur football players showed that such training increased adherence to dietary recommendations by 40%, directly correlating with improved performance metrics [22].

This study underscores the need for further exploration of how dietary patterns influence referees’ physical and psychological performance. For example, while the role of hydration in cognitive function is well-documented, its specific impact on referees’ decision-making remains under-researched. Similarly, long-term studies investigating the correlation between dietary quality and injury rates among referees could provide valuable insights.

Additionally, the role of micronutrients such as iron, magnesium, and zinc in referees’ performance should not be overlooked. These nutrients are critical for muscle function, energy metabolism, and mental clarity. Research on football players found that insufficient magnesium intake was associated with a 14% increase in muscle cramps and fatigue during high-intensity matches. Including these factors in future analyses could help refine dietary guidelines for referees.

The results of the study suggest that the introduction of educational programs, such as meal planning workshops, could help improve the quality of the judges’ diets. These interventions could include practical tips on preparing healthy meals and principles for optimal hydration on the day of competition.

### Strengths and Limitation

The study possesses several strengths that contribute to its robustness and relevance. First, it addresses a significant research gap by focusing on the dietary habits of football referees in Poland, a group often overlooked in sports nutrition studies. The large sample size of 258 referees, representing diverse age groups, professional experience levels, and educational backgrounds, enhances the generalizability of the findings. The use of a validated questionnaire, with a high Cronbach’s alpha value of 0.82, ensures reliability and internal consistency in capturing dietary behaviors. Additionally, the incorporation of statistical analyses to identify patterns and correlations provides a solid foundation for interpreting the data. The ethical adherence to national regulations and the voluntary and anonymous nature of the survey further strengthen the study’s credibility.

However, the research also has limitations that should be acknowledged. The reliance on self-reported data introduces the possibility of recall bias or social desirability bias, which may affect the accuracy of responses regarding dietary habits and behaviors. The cross-sectional design limits the ability to establish causal relationships between dietary practices and professional performance, as it captures data at a single point in time. Furthermore, while the study includes a large sample, it may not fully represent referees officiating at higher professional levels or those from less urbanized areas. The use of an online survey format, though convenient, may have excluded individuals with limited access to digital resources. Lastly, the lack of direct measurements, such as biochemical markers or detailed dietary analyses, restricts the depth of insight into the nutritional adequacy of the referees’ diets and its physiological implications. Future research could address these limitations by incorporating longitudinal designs, objective data collection methods, and a broader demographic representation.

A key limitation of the study is its reliance on declarative data, which may be subject to memory error. Future studies should consider using food diaries and dietary biomarkers, which would allow a more objective assessment of dietary habits.

## 5. Conclusions

Football referees should pay greater attention to meal regularity and the quality of their food choices. The study results indicate that irregular eating schedules and insufficiently healthy cooking techniques may negatively impact the functioning of this professional group. Promoting cooking methods such as steaming or baking could contribute to improving the overall dietary habits of referees.

Attention should also be given to hydration habits, as proper hydration is essential for maintaining both physical and mental performance. Excessive fast food consumption and frequent snacking may pose challenges requiring educational interventions, particularly since a lower proportion of respondents reported choosing whole-grain products or fish in their diet.

Introducing dedicated training programs for referees, especially those at the start of their careers, could significantly enhance their nutritional awareness and lead to improved performance in daily activities and professional duties. These findings also underscore the need for further research into the impact of diet on physical and psychological performance in this professional group.

## Figures and Tables

**Figure 1 nutrients-17-00401-f001:**
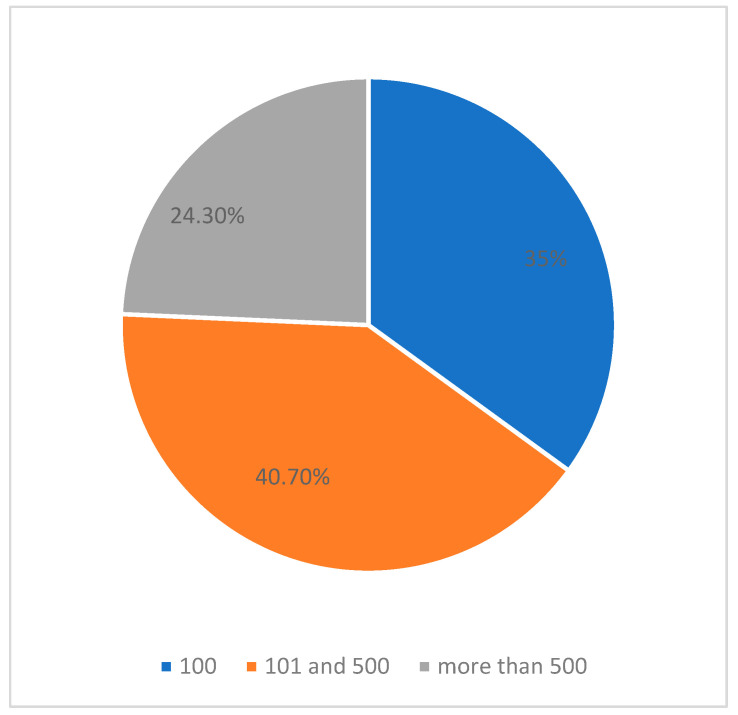
Number of matches officiated per month.

**Figure 2 nutrients-17-00401-f002:**
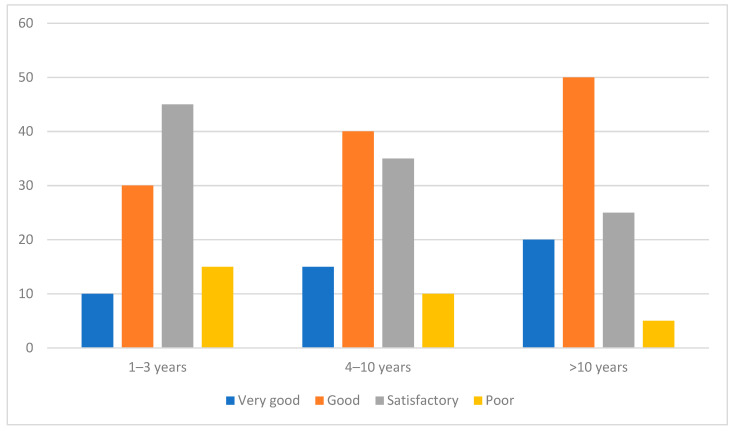
Relationship between work experience and level of food habits (%).

**Table 1 nutrients-17-00401-t001:** Study group characteristics.

Group characteristics	
Gender	
Male	90.4%
Female	9.6%
Age	
<20 or >50 years	8.1%
21–30 years	51.9%
31–40 years	25%
41–50 years	15%
Residence	
Urban	67.4%
Rural	32.6%
Education	
Higher	57%
Secondary or vocational	43%
Professional experience	
1–3 years	31.1%
4–10 years	40.0%
>10 years	28.9%
Number of matches	
<100 matches	35.0%
101–500 matches	40.7%
>500 matches	24.3%
Level of competition	
Amateur leagues	32.6%
Regional leagues	25%
National competitions	42.4%

**Table 2 nutrients-17-00401-t002:** Detailed study findings.

Variable	Comparison	Percentage	*p*-Value
Irregular meal timing	Less experienced vs. more experienced referees	45.2%	<0.05
Fruit consumption	Older (41–50) vs. younger referees	34.1%	<0.01
Fish consumption	Higher education vs. vocational education	60.7%	<0.01
Water intake	High-level competition referees vs. others	28.2%	<0.05
Frying as main cooking method	General population	48.1%	<0.01

**Table 3 nutrients-17-00401-t003:** Relationship between work experience and level of food habits.

Professional Experience	Nutritional Knowledge Level	Percentage	*p*-Value
1–3 years	Very good	10%	*p* < 0.01
	Good	30%	
	Satisfactory	45%	
	Poor	15%	
4–10 years	Very good	15%	
	Good	40%	
	Satisfactory	35%	
	Poor	10%	
>10 years	Very good	20%	
	Good	50%	
	Satisfactory	25%	
	Poor	5%	

## Data Availability

The original contributions presented in the study are included in the article; further inquiries can be directed to the corresponding author.

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
