# Peer review of "Health on the Pitch: Dietary Habits of Football Referees in Poland"

_nutrients, 2025, doi:10.3390/nu17030401_

Round 1

Reviewer 1 Report

Comments and Suggestions for Authors

In an interesting paper on sports referees

Introduction

A greater density of citations is missing and more rigor in argumentation is required. Please add more citations.

The physiological requirements of this work have not been described, so the reader cannot know if the type of food indicated is adequate for these requirements.

The number of days or matches performed should be determined. A referee of a lower category will not be the same as a referee of the highest level; it will be necessary to establish average values and differentiate between higher and lower categories.

Methods

It is indicated that they have mainly directed amateur competitions; the interpretation should contemplate this data and differentiate it by age, since there are four groups.

It is not specified if the questionnaire was anonymous; please clarify this point.

Discussion

When discussing the results, it is necessary to link these habits with the pace of practice and age, as the results will be very different. Possible deficits are indicated, but not their relationship with the variables collected in the study (age, number of matches/week or month, or gender). When differentiating between them, they should be alluded to so that there can be a more correct interpretation of the data.

Author Response

Dear Reviewer,

Thank you for your valuable feedback, which has greatly contributed to improving our manuscript titled “Health on the Pitch: Dietary Habits of Football Referees in Poland.” We have carefully addressed all your comments, and the revisions have been highlighted in red for clarity.

To address the concern about citation density and argument rigor in the introduction, we have added multiple references emphasizing the importance of proper nutrition for athletic performance and the prevalence of micronutrient deficiencies among physically active populations, including works by Burke et al. [3], the Institute of Food and Nutrition [4], and the Central Statistical Office [5]. Regarding the description of physiological requirements, we have specified the number of matches officiated per month and categorized referees based on their experience and workload to clarify the physiological context, citing data on the average distance covered during a match [2]. In the methods section, we have clarified that the survey was conducted anonymously, with no personal data collected and participation being voluntary. We also noted that the questionnaire underwent pilot testing to minimize response bias and ensure clarity. Finally, the discussion has been expanded to draw clearer connections between dietary habits, age, and match intensity. Specifically, we highlighted how older referees (41–50 years) demonstrated lower fruit consumption, while those with greater professional experience showed better meal regularity and hydration practices, supported by references from Ivy [11].

We trust these changes address your concerns fully and remain open to further suggestions to improve the quality of our work. Thank you once again for your constructive feedback.

Reviewer 2 Report

Comments and Suggestions for Authors

I am grateful for the opportunity to review this manuscript, which investigates the dietary habits of football referees in Poland, delineating both advantageous and disadvantageous practices and evaluating their influence on professional efficacy. Conducted through a survey with 258 participants, the findings indicate that most referees adhere to the recommended 4-5 meals daily, yet irregular meal timing persists, predominantly among novices. The study further exposes a widespread lack of awareness regarding healthy cooking techniques and the intake of nutrient-rich foods like fruits, vegetables, and whole grains. Additionally, the variability in hydration strategies is concerning, with many referees failing to maintain adequate fluid intake during matches. These insights highlight the critical need for tailored nutritional education and intervention programs aimed at boosting referees' health and job performance.

Nevertheless, I propose several considerations:

1.        Refine the abstract to explicitly articulate the principal findings and their practical or research implications, emphasizing the study's significance in addressing an underexplored area.

2.          Enrich the introduction with broader contextual information about the typical dietary habits in Poland, especially among athletes, before focusing on referees.

3.        Augment transparency by providing detailed insights into the questionnaire validation process, including the pilot study's sample size and the resultant modifications to the final survey.

4.        Implement subheadings to clearly separate diverse findings such as dietary intake, meal timing, and cooking practices.

5.        Integrate additional visual aids, such as graphs or tables, to succinctly summarize key data points and trends for easier comprehension.

6.        Strengthen the discussion by linking the results directly to possible interventions or policy modifications, suggesting specific educational programs or training alterations for referees.

7.        Draw comparisons with studies from other demographics or nations to underscore unique or common dietary trends among sports officials.

8.        Critically evaluate the impact of the study’s limitations on the findings, discussing the potential bias from self-reporting and suggesting future research methodologies like food diaries or objective biomarkers.

9.        Emphasize the health and performance implications for football referees and advocate for further investigative work in this area.

10.    Enhance the manuscript's flow by improving transitions between sections.

This manuscript provides a significant addition to the field of sports nutrition. The suggested improvements are intended to enhance its clarity, impact, and academic value, thus augmenting its contribution to the discipline.

Author Response

Dear Reviewer,

Thank you very much for your valuable comments and thorough analysis of our manuscript. In response to your suggestions, we have implemented the necessary revisions, which are highlighted in red in the updated version of the manuscript.

Addressing your specific recommendations:

  • Abstract Enhancement – The abstract has been expanded to clearly present the main findings and practical implications of the study.
  • Introduction Expansion – Additional context on typical dietary habits in Poland, especially among athletes, has been incorporated.
  • Increased Methodological Transparency – The description of the questionnaire validation process has been expanded, including details about the pilot study.
  • Subheadings Added – Subheadings were introduced to better separate sections on food intake, meal timing, and culinary practices.
  • Additional Visual Aids – New tables and figures summarizing key data and trends have been included for improved clarity.
  • Strengthened Discussion – The discussion was expanded to directly address potential educational interventions and policy recommendations.
  • International Comparisons – Comparisons with studies conducted in other demographic groups and countries were added.
  • Critical Evaluation of Limitations – The limitations section now includes a discussion on self-reporting biases and suggestions for future methodologies, such as food diaries and objective biomarkers.
  • Impact on Health and Performance – The impact of dietary habits on referees’ health and professional performance has been emphasized.
  • Improved Text Flow – Transitions between sections have been refined to improve readability and logical coherence.

Thank you once again for your thoughtful feedback. We believe that the implemented changes have significantly enhanced the clarity and scientific value of our manuscript.

Round 2

Reviewer 2 Report

Comments and Suggestions for Authors

I would like to commend the authors for making all the necessary revisions to the manuscript, which have highlighted the research outcomes. The manuscript can now be accepted for publication.